



# Development of multi-channel whole-air sampling equipment onboard unmanned aerial vehicle for investigating VOCs vertical distribution in the planetary boundary layer

Suding Yang[1], Xin Li[1,2,3], Limin Zeng[1,2,3], Xuena Yu[1], Ying Liu[1], Sihua Lu[1], Xiaofeng Huang[4], Dongmei Zhang[5], Haibin Xu[5], Shuchen Lin[5], Jinhui Cui[1], Lifan Wang[1], Ying Chen[1], Wenjie Wang[1], Mengdi Song[1], Liuwei Kong[1], Yi Liu[1], Linhui Wei[6], Xianwu Zhu[6], Yuanhang Zhang[1,2,3]

[1]State Key Joint Laboratory of Environmental Simulation and Pollution Control, College of Environmental Sciences and Engineering, Peking University, Beijing, 100871, China
[2]International Joint Laboratory for Regional Pollution Control, Ministry of Education, Beijing, 100816, China
[3]Collaborative Innovation Centre of Atmospheric Environment and Equipment Technology, Nanjing University of Information Science and Technology, Nanjing, 210044, China
[4]Laboratory of Atmospheric Observation Supersite, School of Environment and Energy, Peking University Shenzhen Graduate School, Shenzhen, 518055, China
[5]Quadrant Space (Tianjin) Technology Co., Ltd, Tianjin, 301700, China
[6]Bescient Technologies Co., Ltd, Shenzhen, 518004, China

*Correspondence to:* Xin Li (li_xin@pku.edu.cn)

**Abstract.** To achieve near-continuous vertical observations of volatile organic compounds (VOCs) in the planetary boundary layer (PBL), multi-channel whole-air sampling equipment onboard an unmanned aerial vehicle (UAV) platform was developed in this study. The equipment consists of a multi-position solenoid valve and specially designed lightweight quartz sampling canisters. The canisters have little adsorption loss of VOCs and good inter-canister reproducibility. The 7-day recovery test shows that most VOC species (97%) had a one-week decay within 20%. Online instruments for measuring $O_3$, $NO_2$, CO, $SO_2$, and meteorological parameters are also integrated into the UAV platform. During one take-off and landing, the UAV platform can reach 800 m above the ground within 40 min and take whole-air samples at six heights. Vertical profiles of VOCs and trace gases during the evolution of the PBL in southwest China are successfully obtained by deploying the newly developed UAV system.

## 1 Introduction

Volatile organic compounds (VOCs) play a key role in air pollution via atmospheric oxidation (Li et al., 2020a; Li et al., 2019), and have large spatiotemporal variability owing to the large variety and wide sources of VOCs (Bari and Kindzierski, 2018; Li et al., 2019). Current research is no longer limited to surface observations, but has gradually expanded to aerial surveys to observe the characteristics of VOCs from a three-dimensional perspective (Mo et al., 2018; Sangiorgi et al., 2011; Sun et al., 2018; Vo et al., 2018). Vertical detection of VOCs is essential for advancing our understanding of the distribution and chemical behaviour of VOCs in the planetary boundary layer (PBL), as well as for testing and improving VOCs emission inventory



models (Mo et al., 2022; Li et al., 2017; Wöhrnschimmel et al., 2006; Velasco et al., 2008). However, the existing technology of VOC aerial observation has deficiencies, which hinder our in-depth understanding of the vertical distribution of VOCs in

the PBL.

The vertical profile of VOCs can be measured using different technologies, such as tower observations, aircraft, tethered balloons, and unmanned aerial vehicles (UAV). Tower measurements can perform long-term continuous observations, but only within a few hundred meters because of height limitations (Mo et al., 2020; Li et al., 2020b; Qiu et al., 2019). Aircrafts can measure on a scale of several kilometres, but this method is time-consuming and unable to work in the near-ground atmosphere

(Nunnermacker et al., 2008; Geng et al., 2009; Benish et al., 2020). Tethered balloons lack stability and manoeuvrability (Wu et al., 2020; Sun et al., 2018). The advantages of miniaturisation and manoeuvrability of UAVs may overcome the deficiencies of current observation technology and realise mesoscale observations with higher spatial-temporal resolution in the boundary layer (Vo et al., 2018; Asher et al., 2021).

The UAV platform has been applied to a series of observations of biogenic VOCs (BVOCs) in the Amazon rainforest

(McKinney et al., 2019; Batista et al., 2019; Guimarães et al., 2019). Cartridges loaded with Tenax TA and Carbograph 5TD sorbents were mounted on a hexacopter to collect gas- and aerosol-phase VOCs for subsequent offline analysis by thermal desorption gas chromatography-mass spectrometry (TD-GC-MS). This method enables the measurement of samples collected continuously for 10 min, with an overall minimum detection limit of 3 ppt. However, the types of target compounds for sorbent sampling are limited, and volatile compounds may also have problems with penetration loss (Li et al., 2021b; McKinney et al.,

2019). Canister sampling enables the full-component sampling of air, avoiding penetration decomposition and loss of adsorbent sampling.

The SUMMA polished canister, which is passivated into a smooth nickel-chromium oxide surface to reduce the internal surface area in contact with air samples, has been widely used to collect air samples of VOCs (Ochiai et al., 2002; Pang et al., 2015; Shao et al., 2009). Previous studies have illustrated the use of SUMMA canisters mounted on a multi-wing UAV platform for

whole-air sample collection of VOCs (Liu et al., 2021; Chang et al., 2016). The sampler consists of a separate vacuum canister and a remote-control valve. Owing to the limited load capacity of the UAV, only one SUMMA canister can be carried in a flight mission, making continuous gradient measurement in vertical observation difficult to achieve. The development of sampling canisters suitable for aircraft observation has become a critical task; canisters should be lightweight and small, with effective collection and storage of VOC air samples.

In this study, a new whole-air quartz sampling canister is developed and characterised. Benefiting from its light weight, six canister samples can be taken during one flight mission, which enables capturing the vertical distribution of VOCs with better time resolution compared to previous UAV observations. In addition, we integrate the measurements of O3, NO2, SO2, CO, relative humidity, temperature, and pressure on the same UAV platform, which is conducive to an in-depth understanding of photochemical reaction progress in the upper atmosphere. We show that the newly designed UAV system can be successfully

deployed to investigate the evolution of vertical profiles of VOCs and other trace gases during PBL development.



## 2 Materials and methods

### 2.1 Flight platform

#### 2.1.1 UAV configuration

The UAV platform used in this study was an AX-50J hexacopter (Quadrant Space (Tianjin) Technology Co. Ltd., Tianjin,
China). The centre frame had a width of 1.83 m and a height of 0.81 m. The weight of the UAV itself was 15.4 kg with a
maximum payload of 22 kg. The maximum climbing speed of the UAV was 5 m/s. Flight could be performed under strong
breeze conditions (wind speed ≤13.8 m/s) and at an ambient temperature of −10 to 40 °C. The UAV was powered by two high-
density lithium polymer batteries (1332 Wh, 50 V), which could provide a flight time of 43 min under the maximum payload.
The actual battery usage during each flight depended on the flight pattern, wind speed, and air temperature. Our test flights in
winter (ambient temperature was approximately 0–4 °C and wind speed was below 2 m/s) showed that the UAV with full
payload could reach 800 m above ground. The take-off and landing usually took 30 min. The maximum battery consumption
was 922 Wh, which was less than 70% of the total battery capacity.

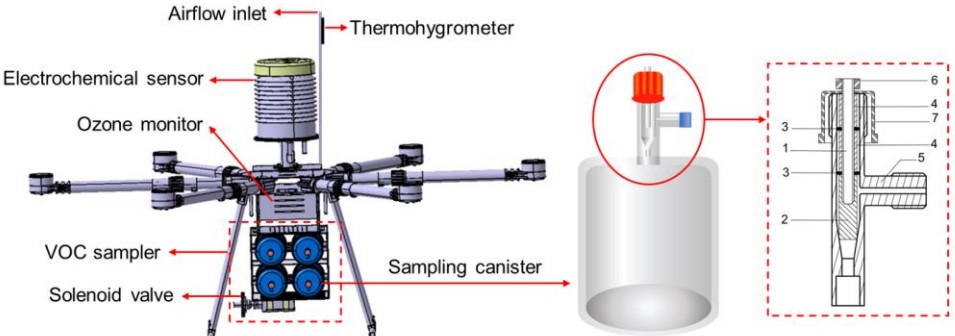

**Figure 1. Left: the UAV sampling system. Right: schematic diagram of designed quartz sampling canister (1: stainless-steel column**
**with internal thread; 2: polytetrafluoroethylene column; 3: rubber O-ring; 4: glass valve with inner inclined cone; 5: air inlet channel;**
**6: stop-screw nut; 7: screwable part).**

#### 2.1.2 Control system

The UAV platform was equipped with a control module for flight tasks and safety issues. An integrated flight controller with
a high-precision global positioning system (GPS) and inertial measurement unit (IMU) could execute flight or hover commands
at a specified location and maintain interactive communication with a handheld-control ground station. The positioning
accuracy was ± 0.5 m in the vertical direction and ± 1 m in the horizontal direction. A KX control computer was used in the
navigation control system to transmit airborne and ground signals through the radio (operating frequency: 2.4 GHz). The
onboard camera and ground station screen could display real-time images and flight data (such as real-time altitude, longitude
and latitude, flight speed, and power supply voltage) for power monitoring and timely diagnosis of motor problems. The multi-
copter could be set to autonomous flight mode by editing the flight path and hover time using the ground station program, or



it could be manually controlled by the ground station during flight. An automatic return program was developed to ensure safe return of the UAV to the ground in situations of insufficient battery power or signal loss.

## 2.2 Sampler description

### 2.2.1 The whole-air quartz sampling canister

As shown in Figure 1, this study developed a new type of sampling canister composed of a quartz cylindrical tank and an airtight valve. The canister was 14.3 cm high, 9.8 cm in diameter, 0.7 cm in wall thickness, approximately 1.1 L in volume, and approximately 415 g in weight. The cylindrical canister body facilitated rapid installation while ensuring a small surface area to reduce the wall effect. The airtight valve consisted of seven parts, as shown in Fig. 1. The stainless-steel column with internal thread was fixed using a stop-screw nut with a screwable red cap, and the lower part was combined with a

polytetrafluoroethylene (PTFE) column. Screwing of the red cap could control whether the PTFE column was in contact with the inner surface of the glass valve. Two rubber O-rings ensured the airtightness of the valve. Before sampling, the quartz canister was purged with nitrogen and depressurised to a near-vacuum state (30 mTorr) using an Entech 3100D canister cleaning system. During sampling, the PTFE column was separated from the inner surface of the valve. The gas channel, where the blue cover is shown in Fig. 1, connected the sampling tube and solenoid valve. When the corresponding valve was opened,

ambient air was sucked into the canister through the channel. Appropriate sampling times were required to ensure that the sample volume met the analytical demand (see Section 3.1.4). The inner surface of the canister was polished to minimise the number of active adsorption sites and reduce the wall effects. To avoid sample loss due to photochemical reactions, we created a light-proof box for each canister to store the samples. The sampling canisters were confirmed to be suitable for UAV sampling and subsequent offline analysis through a series of characterisation experiments (see Section 3.1 for more information).

**Table 1. Measurement parameters and instrument information in this study.**

| Parameter | Instrument | Measure method | Resolution | Accuracy | Measurement range | Weight | Time interval |
|---|---|---|---|---|---|---|---|
| VOCs | VOC sampler | offline sampling | - | - | - | 3.12 kg | - |
| O$_3$ | O$_3$ monitor | online | 0.1 ppb | ±1.5% | 0–250 ppm | 1.46 kg | 2s |
| NO$_2$ | electrochemical sensor | online | 4 ppb | ±2% | 0–20 ppm | 5.15 kg | 5s |
| CO | | | 2 ppb | ±2% | 0–1000 ppm | | |
| SO$_2$ | | | 3 ppb | ±2% | 0–100 ppm | | |
| P | | | 0.01 hPa | ±1.5% | 30–120 kPa | | |
| T | thermohygrometer | online | 0.1 °C | ±0.5 °C | -10–60 °C | 0.34 kg | 1s |
| RH | | | 0.1% | ±2.5% | 0–100% | | |

### 2.2.2 Ancillary instruments

Ancillary observation instruments included an ozone monitor, electrochemical sensors, and a thermo-hygrometer. Table 1 shows detailed information of the instruments. The measuring principle of the ozone monitor (Model 205, 2B Technologies, USA) was ultraviolet absorption. The ozone-scrubbed and unscrubbed air alternately passed through the two separate



absorption cells. Ozone was measured based on the attenuation of light passing through the absorption cells, and its concentration was calculated according to the Beer-Lambert law. This instrument has been widely used in previous aircraft aerial survey research (Chen et al., 2020; Sagona et al., 2018; Wang et al., 2017), showing reliable stability and observational results. The ozone monitor was calibrated using an ozone calibrator before the measurement. Electrochemical sensors (AQMS-6000S, ZTE instruments (Shenzhen) Co. Ltd., China) were used to measure SO2, NO2, and CO, and a built-in barometer was

used to measure air pressure. Electrochemical sensors generate a current proportional to the gas concentration by reacting with the gas molecules. Measurement accuracy has been proven in previous studies (Cross et al., 2017; Mueller et al., 2017). However, the measurement results may be affected by changes in temperature, humidity, and pressure during the flight; therefore, it was necessary to conduct data calibration based on experimental conditions (see Section 3.2.2 for details). Meteorological parameters, including ambient temperature (T) and relative humidity (RH), were measured using a miniature

thermo-hygrometer (Testo 176-H1, Germany).

### 2.2.3 Integration of the sampling equipment

The integrated flight device included a UAV platform, multi-channel VOC sampler, ancillary observation loads, and power and control system. The battery supplied power to the components of different voltages through the voltage divider. The multi-channel VOCs sampler consisted of six or eight evacuated whole-air sampling canisters and a 16-position solenoid valve (Valco

Instruments Co. Inc., USA). The sampling canisters were inserted horizontally into the bracket made of light-proof sponge material, and each bracket was connected to the even-numbered position of the solenoid valve through a Teflon tube. Position 1 of the solenoid valve was connected to the ambient atmosphere through a sampling tube, and the other odd-numbered valve positions were closed with screws. The solenoid valve was actuated by a separate remote-control unit, independent of the UAV controller. When switching to the even-numbered valve position, sampling started; when switching to the odd-numbered valve

position, sampling stopped. The VOC sampler was fixed underneath the multi-copter by straps, which enhanced the centre of gravity to increase stability and facilitate quick disassembly and installation. Although mounting on the top of the multi-copter could reduce the dead volume of the sampling tube, it was observed that the temperature of the top surface became extremely high (approximately 40 °C) owing to direct solar radiation in summer, which may be detrimental to the chemical stability of the VOCs samples.

We referred to the tuft method in a previous study (Zhu et al., 2019) to determine the position of the sampling inlet, which was the minimum height above the rotors without a rotating disturbance. The results showed that there was little influence at 0.5 m above the rotors, similar to the airflow simulation results of the UAV in a previous study (Mckinney et al., 2019). In this study, a support rod fixed the sampling inlet 1 m above the fuselage. A particulate filter membrane was then placed in the sampling tube. To reduce the dead volume, sampling tubes with small inner diameters (3.18 mm) were selected for this study.

The dead volume was calculated as approximately 3.1 mL, only 0.8% of the canister. During the field measurement, the UAV flew to the highest position (800 m) after taking off and hovering at the planned height for sampling during descent. This flight pattern reduced the effect of the dead volume on the sampling at the next altitude (generally, the VOC concentration was lower



at high altitudes) and was conducive to flight safety.

The ozone monitor and electrochemical sensor were placed on top of the UAV platform, which performed real-time
measurements simultaneously with VOCs sampling. Top mounting allowed for a faster temporal response and higher spatial
resolution. A custom-made carbon fibre box suitable for ozone monitoring provided thermal insulation and reduced vibration
interference. The sampling inlet of the ozone monitor was fixed at the same position as the VOC sampling inlet, and a thermos-
hygrometer was installed here.

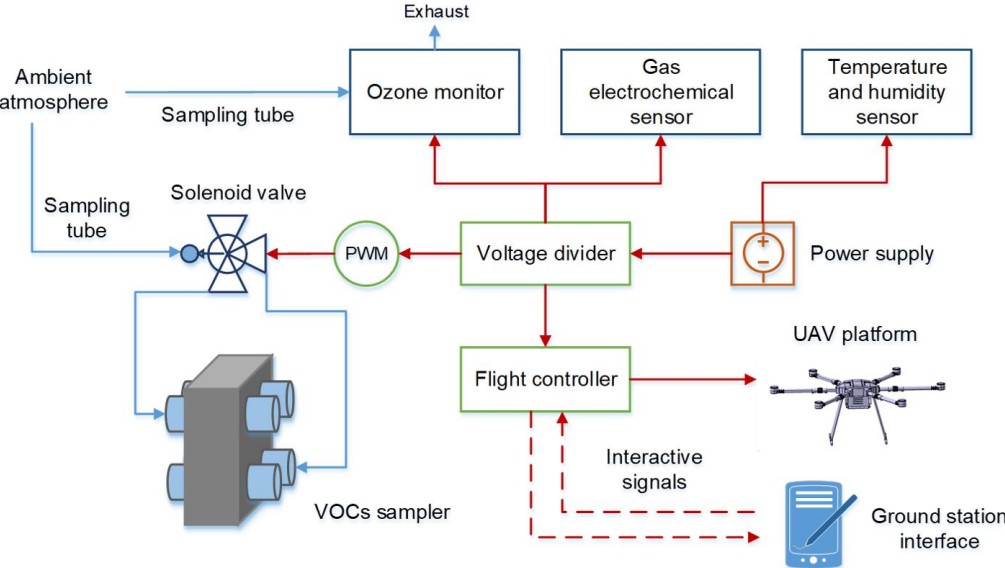

**Figure 2. Schematic diagram of integrated sampling device (the blue and red lines represent air flow and electronic connections, respectively).**

## 2.3 Offline analysis of VOC sample

The collected samples were analysed within one week in the laboratory. An offline gas chromatography–mass
spectrometry/flame ionisation detector (GC-MS/FID) system (Wuhan Tianhong Instrument Co. Ltd., China) was used to
analyse 97 VOC species, including 30 alkanes, 10 alkenes, acetylene, 16 aromatics, 26 halohydrocarbons, and 14 oxygenated
volatile organic compounds (OVOCs). The detailed analysis procedures have been described in previous studies (Yang et al.,
2021; Song et al., 2021). Briefly, the samples collected in the canisters were drawn into a cryoconcentration trap at −160 °C
after removal of $CO_2$ and moisture. Then, the trap was rapidly heated to 110 °C for thermal desorption and injected into the
GC system using a high-purity helium carrier gas. C2–C5 hydrocarbons were separated using a PLOT-AI2O3 chromatographic
column (15 m × 0.32 mm ID × 3 μm, J&W Scientific, USA) and measured by FID. The remaining heavier VOCs were
separated on a DB-624 column (60 m × 0.25 mm ID × 1.4 μm, J&W Scientific, USA) and connected to MS for measurement.
The calibration curve linearity ($R^2$ value) for all target species was greater than 0.99. Further details are provided in table S1.



## 3 Result and discussion

### 3.1 Performance test of the VOC sampler

The VOC sampler adopted the designed quartz sampling canisters, and performance characterisation experiments were
conducted. The advantage of canister sampling is that it ensures the integrity of the sample and facilitates subsequent
analysis, because one sample can be injected several times. Nevertheless, the performance of the new quartz canister must be
investigated to ensure that the measurement results are as close as possible to the actual sampled atmosphere. This study
examined the possible adsorption loss of the sampler, inter-canister reproducibility, stability of the stored samples, and
required sampling time. Commercialised SUMMA canisters were used as experimental controls. The test results demonstrate
the advantages and availability of the VOC sampler proposed in this study.

### 3.1.1 Adsorption loss test

We examined the possible adsorption losses of target compounds during the entire sampling process, including a solenoid
valve, connector with a particulate filter, Teflon tubes, and quartz sampling canister, by comparing the measurement results of
direct injection and passing through the VOC sampler. Before the experiment, the entire sampling line was flushed with high-
purity nitrogen for 5 min. Direct injection connected the airflow outlet of the standard gas dilution system to the pre-
concentration inlet bypassing the VOC sampler. In the comparison group, the standard gas was fed into the VOC sampler, and
the quartz canister with the standard gas was removed from the sampler and connected to the GC-MS/FID for subsequent
analysis. In this study, a standard gas of 2 ppbv was used, and the average value was obtained from three measurements in
each group of experiments. The measurement results for all the VOC species are shown in Table S2. OVOCs were the VOC
group with the highest loss, with a propanal loss of 4.4%. The results of direct injection and passing through the sampler were
comparable, with differences within 3% for most VOC species (96%), indicating negligible adsorption loss of analytes.

### 3.1.2 Inter-canister reproducibility

To test the reproducibility between canisters, six quartz canisters (10% of the total) were randomly selected and measured with
the same concentration of standard gas (1 ppbv), and the SUMMA measurement results were taken as a control. The ratio of
the peak area of each VOC species to the internal standard compound was used as the evaluation reference, which offset the
error caused by instrument fluctuations. In Fig. 3, the box plot shows the experimental results of quartz canisters, with the
average values of SUMMA canisters marked separately with red plus signs. The results suggested good inter-canister
reproducibility.

We used the F-test to check the variance homogeneity of the quartz and SUMMA canister measurements. The results showed
no significant difference ($p<0.05$), indicating that the measurement precision of the quartz canister was equivalent to that of
SUMMA. The reproducibility of the measurements for each VOC species was calculated using the coefficient of variation
(CV). For all VOC species, the CV ranged from 1.3% to 19.0%. The reproducibility of the measurements varied between





species. The average CV of alkanes, alkenes, acetylene, aromatics, OVOCs, and halohydrocarbons were 5.8%, 4.5%, 3.1%, 9.3%, 7.1%, and 6.2%, respectively. The CV of high-carbon alkanes, aromatics, and certain OVOCs was high, which included the differences between canisters and the uncertainty in the analysis process. Uncertainties existed in the instrument's measurement of high-carbon hydrocarbons and multifunctional compounds, which might amplify errors in the results. In general, the average CV of the quartz canister measurement was 6.9%, which was slightly lower than the 7.2% for SUMMA, meeting the VOC measurement accuracy requirement.

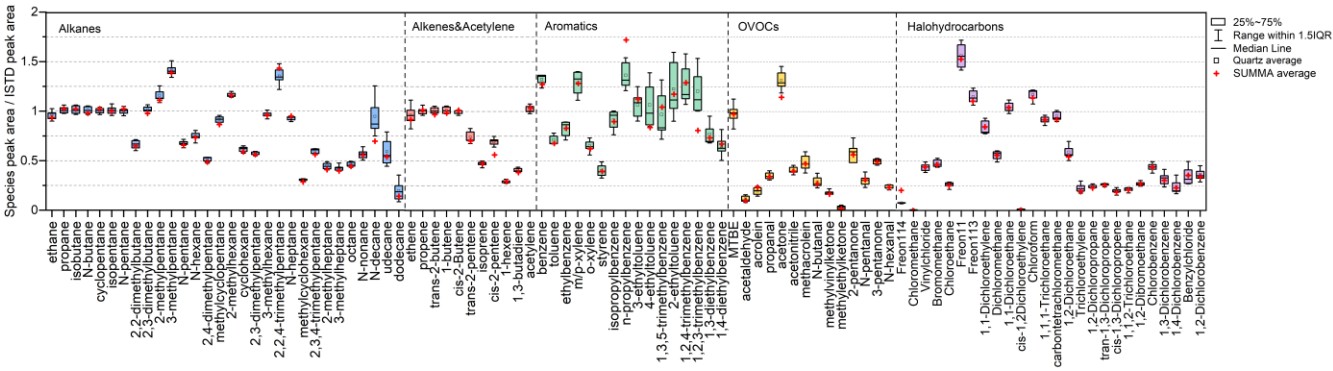

**Figure 3. Result of the inter-canister reproducibility experiment.**

### 3.1.3 Canister stability

To test the stability of VOC samples stored in canisters, the 7-day recovery rate was used to evaluate the stability, as it was the time normally required to complete the collection, transportation, and analysis of samples. Four quartz and four SUMMA canisters were used to collect ambient gas through the same sampling inlet by a cross joint, which kept the initial concentration of VOCs in each canister consistent. Ambient samples were collected at noon, with high concentrations of ozone (83 ppbv) and a relative humidity of 40%. In addition, quartz canisters with 2 ppbv standard gas were used as controls. Measurements were performed at different time intervals (0, 3, 5, and 7 d), and each sample was measured twice to obtain the average results. Table S2 shows the measurement results and 7-day recovery rate for each group of experiments. For ambient samples, the recoveries of the VOC species in the quartz and SUMMA canisters ranged from 76.5% to 110.0% and 77.2% to 111.1%, respectively. Some species with low chemical reactivity, such as low-carbon alkanes, aromatics, acetylene, and halohydrocarbons, had a relatively high 7-day recovery (>90%). In general, the average VOC recovery rate in quartz canisters was 94.91%, which was slightly higher than that in SUMMA canisters (94.37%), indicating a stable storage environment in quartz canisters. The recovery rate of the ambient samples was generally lower than that of standard gas samples (97.3%). This could be due to concentration attenuation caused by the chemical reaction of oxidants, such as ozone, with VOCs. However, this impact could be ignored because 97% of the VOC components beyond the detection limit decreased by less than 20% after being placed in the quartz canister for a week.



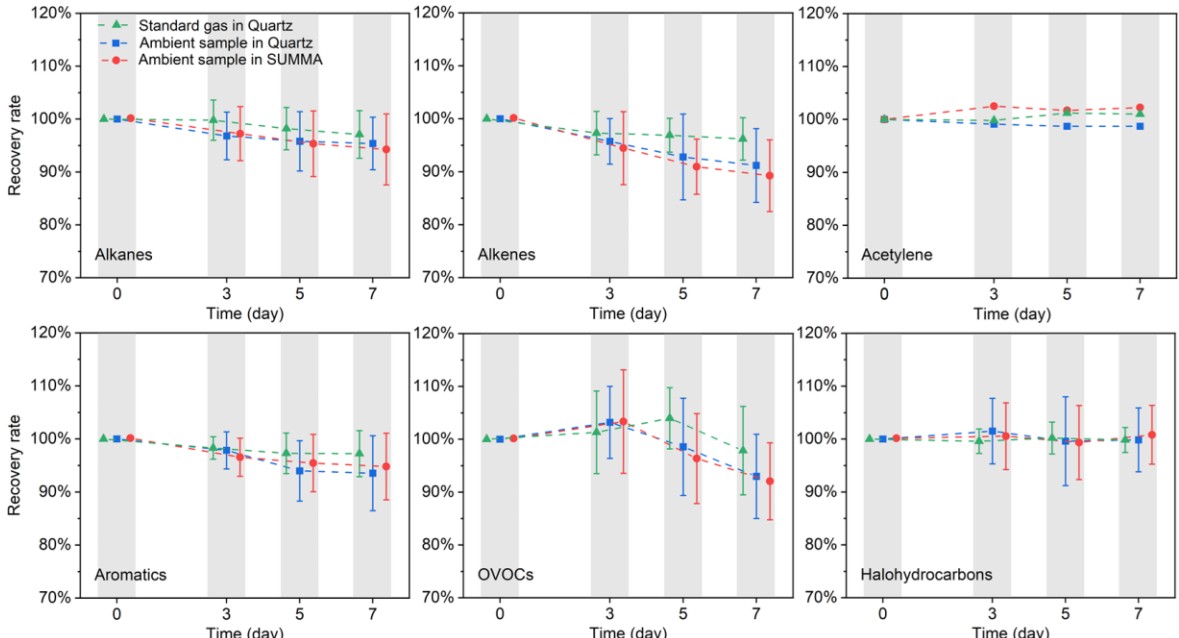

**Figure 4. 7-day recovery rate for different VOC groups.**

Figure 4 shows the trend of the recovery rate over time for the different VOC groups. The decay trends of each VOC group of ambient samples in the quartz and SUMMA canisters were similar: the concentration of OVOCs first increased and then decreased with time, and other VOC species decreased with time. The concentration of the standard gas sample changed slightly over time. For ambient samples, the 7-day recoveries of alkanes, alkenes, acetylene, aromatics, OVOCs, and halohydrocarbons in the quartz canisters were 95.3%, 91.2%, 98.4%, 93.5%, 91.9%, and 99.6%, respectively. The average recoveries of alkanes, alkenes, and OVOCs in quartz canisters were slightly higher than those in SUMMA canisters. However, it is worth noting that some alkenes, such as 1-hexene and cis-2-pentene, decreased significantly. Timely analysis after sampling (e.g., within 3 d) could effectively reduce this attenuation. Therefore, the samples in this study were analysed within 3 d of sampling to ensure the highest possible recovery rate.

### 3.1.4 Determination of the sampling time

There is a trade-off between sampling time and quantity of samples per flight. Owing to battery limitations, we attempted to collect more samples in the shortest possible time. The sampling time required for the quartz canister was first determined on the ground. To verify the airtightness of the valve, the canister depressurised to near vacuum (0.03 Torr) was attached to the valve. The internal air pressure was measured after 24 h and remained constant. The valve was opened for sampling at intervals of 60, 75, 90, and 120 s. The results showed that the pressure in the tank reached equilibrium with the ambient pressure (14.7 psi) after 90 s. Owing to the lower air pressure at high altitude, the time required for the sampling canister to reach equilibrium could be different from that on the ground. To determine the appropriate sampling time, hovering at different time intervals





was performed at the highest flight altitude of 800 m. The correspondence between the sampler and the user interface allowed remote monitoring and valve switching during flight. The results indicated that 2 min was the most suitable sampling time.

## 3.2 Performance test of ancillary observation instruments

### 3.2.1 Calibration and comparison experiment

The ozone monitor was calibrated using an $O_3$ calibration source (Model 49i Primary Standard, Thermo Fisher Scientific, USA) before being used on the UAV platform. The $O_3$ calibrator can generate airflow with a set $O_3$ mixing ratio (user-adjustable from 0 to 1000 ppb). A sampling tube was used to connect the sampling inlet of the ozone monitor to the outlet of the $O_3$ calibrator to conduct the calibration. Five target $O_3$ mixing ratios in the range 0–100 ppb were used in the process. We calculated the
slope and intercept of the linear fit equation to calibrate the instrument based on the results of the multipoint measurements (Fig. S1). Because the physical characteristics of ozone molecules hardly change in the PBL, the calibration factor of the $O_3$ monitor measured based on UV absorption showed little change at different altitudes. In this study, only $O_3$ monitor laboratory calibration was conducted, and the calibration data of the electrochemical sensor were provided by the manufacturer.

To further determine the accuracy of the instrument measurements, comparison experiments with conventional regulatory
monitors were conducted at the Chengdu National Environmental Monitoring Station from 27 to 28 December 2021. The regulatory monitors were of the Thermo Fisher series, and the relevant information is listed in Table S3. The instruments were installed near the sampling inlet of the regulatory monitor on the roof of the station to ensure measurement of the same air mass. Figure S2 shows a time-series comparison of the measurements for each pollutant. For all pollutants, the instruments used in this study and the regulatory monitors showed consistent trends in the time series. However, the measurements of the
ozone monitor and electrochemical sensor showed larger fluctuations than those of the regulatory monitors, possibly because of the shorter time intervals. In the actual measurements, we took the average value of the effective time as the result. The comparison experiment proved the reliability of the measurement results of the ancillary observation instrument at ground level.

### 3.2.2 Influence of environmental factors (T, RH, P) on measurement results

Although ancillary observation instruments obtained relatively accurate results on the surface, they could be affected by variations in ambient environmental factors (such as temperature, relative humidity, and pressure) during flight. Ozone monitors based on UV absorption have been widely used in aerial research (Chen et al., 2020; Sagona et al., 2018; Wang et al., 2017). Researchers have found that temperature is an important factor affecting the normal operation of ozone monitors, and an insulation box can solve this problem effectively (Chen et al., 2020; Li et al., 2020). We designed a carbon-fibre insulation
box for the ozone monitor with a sponge on its inner surface. The ozone monitor with/without insulation box was placed on the UAV platform to conduct the measurements at an altitude of 200 m in high-temperature weather (surface temperature 36 °C). Figure S3 shows the measurement results for the two situations, where the insulation box reduced the data fluctuation





of the ozone monitor during the vertical observation. The standard deviation of the measurement results using the insulation box (±2.64 ppbv) was smaller than that without (±5.13 ppbv), indicating that the measurement stability of the ozone monitor was improved.

For electrochemical gas sensors, it is necessary to understand the influence of environmental factors on the working electrodes of the sensors. A previous study indicated that electrochemical sensors are sensitive to changes in ambient temperature (Cross et al., 2017). This study tested the signal response of the sensor unit over a temperature range of −30 °C to 50 °C. Figure S4 (a) shows the experimental results and fitting equations for the three gas sensors. In the actual measurement, the fitted nonlinear equation was programmed into the data processing code to correct for the effect of temperature variation. Figure S4 (b) shows the influence of temperature on the sensor null point, which remained constant (change less than 3 ppbv) in the temperature range of −10 °C to 30 °C. According to the information provided by the manufacturer, the air pressure changes in the PBL were within the normal working pressure (80–120 kPa), so it would not interfere with the sensor's normal operation. Research suggests that changes in relative humidity would cause current peaks but do not affect the final measurement results (Mueller et al., 2017). This study found that the signal interference caused by an RH mutation of 10% (the maximum RH change per 100 m in the experiment) returned to the initial state within 30 s. In actual observations, the RH at a certain altitude remained almost unchanged with time (see Fig. 5), so hovering for enough time during the measurement and carefully selecting effective data could avoid the interference caused by humidity changes.

### 3.3 Field experiments

To test the availability of the whole-air sampling equipment, field observation experiments were conducted in Chengdu, a city in southern China, from 22 December 2021 to 5 January 2022. The sampling site was located west of the Chengdu Plain (30.66 °N, 103.65 °E, altitude: 527 m). There were some motor vehicles, but no other significant sources of emissions. The execution times for each VOC sampling mission are listed in Table S4. A total of 182 VOC samples (26 vertical profiles) were obtained during observation. All vertical profiles were acquired at six gradients (800, 650, 500, 350, 200, and 100 m), hovering at each height for 2 min for VOC sampling and stabilising instrument measurement. VOC samples were initially collected manually from the ground. Before each flight, the ozone monitor and electrochemical sensor were turned on and warmed for at least 20 min to obtain stable readings.

Figure 5 shows an example of the time series of the measured parameters and the corresponding flight heights during a mission, the first flight on the morning of December 29th. As shown in the figure, the temperature inversion occurred between approximately 100 m and 200 m near the ground, with obvious stratification in the relative humidity and air pressure at different heights. The variation trend of O3 was different from those of CO and NO2. O3 concentrations showed a positive gradient with altitude in the morning owing to the combined effects of inhibited vertical mixing, NO titration, and surface dry deposition. This is consistent with the high concentrations of O3 observed in the nocturnal residual layer in other studies (Chen et al., 2020; Li et al., 2021). The concentrations of NO2 and CO decreased with altitude, with little NO2 above 500 m. The SO2 concentration was below the detection limit during measurement. During flight, external factors might cause fluctuations in



instrument measurements, which decreased after reaching a certain target altitude for a period. We considered the data valid if the coefficient of variation of the results at each target height was within 20%. VOCs accumulated near the surface under the inversion layer, with a mixing ratio of 57.43 ppbv. For the composition, alkenes and aromatics with high reactivity decreased sharply above 100 m, indicating that the low boundary layer height in the morning inhibited the vertical transmission of freshly emitted VOCs.

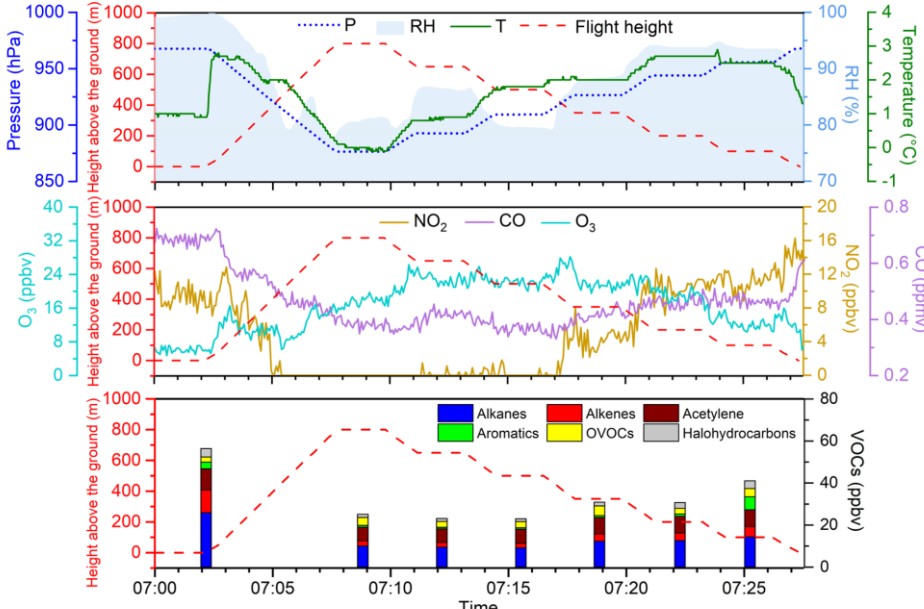

**Figure 5. Example of the time series of measured parameters and the corresponding flight heights during a mission.**

To study the vertical distribution evolution of VOCs during PBL development, we conducted flight missions four times per day: (1) stable boundary layer (before sunrise, LT 07:00–7:30), (2) during PBL development (in the morning, LT 10:30–11:30), (3) PBL fully opened (in the afternoon, LT 14:00–15:00), and (4) after boundary layer descent (in the evening, LT 19:00–20:00). The boundary layer height data were obtained from ERA5 data (the fifth generation ECMWF atmospheric reanalysis of the global climate). As shown in Figure 6, from the evening of 28 December to the early morning of 29 December the PBL height was low, and VOCs accumulated on the surface. The VOC concentration and chemical composition remained unchanged above 200 m, as the residual layer retained the VOCs of the previous day. Similar phenomena also occurred in the early mornings of other sampling days. After sunrise, thermal convection enhanced the upward transport of VOCs from the surface, reaching the maximum concentration of 69.49 ppbv at 100 m. In the afternoon, VOCs were well-mixed within the fully opened PBL, and uniform vertical distribution patterns were observed below 800 m. The concentration of OVOCs increased at higher altitudes with a larger proportion. VOCs on the surface increased cumulatively after the boundary layer descended below 100 m, from 47.77 ppbv in the afternoon to 56.76 ppbv. The results of this study can provide additional insights into the spatial variation and evolution patterns of VOCs in the winter boundary layer. In previous studies, it took





nearly two hours to obtain a complete VOC profile using single-canister sampling (Li et al., 2021), during which the PBL was likely to develop and change rapidly. The newly designed UAV system in this study can perform sampling within half an hour to capture snapshots of the VOC vertical variability.

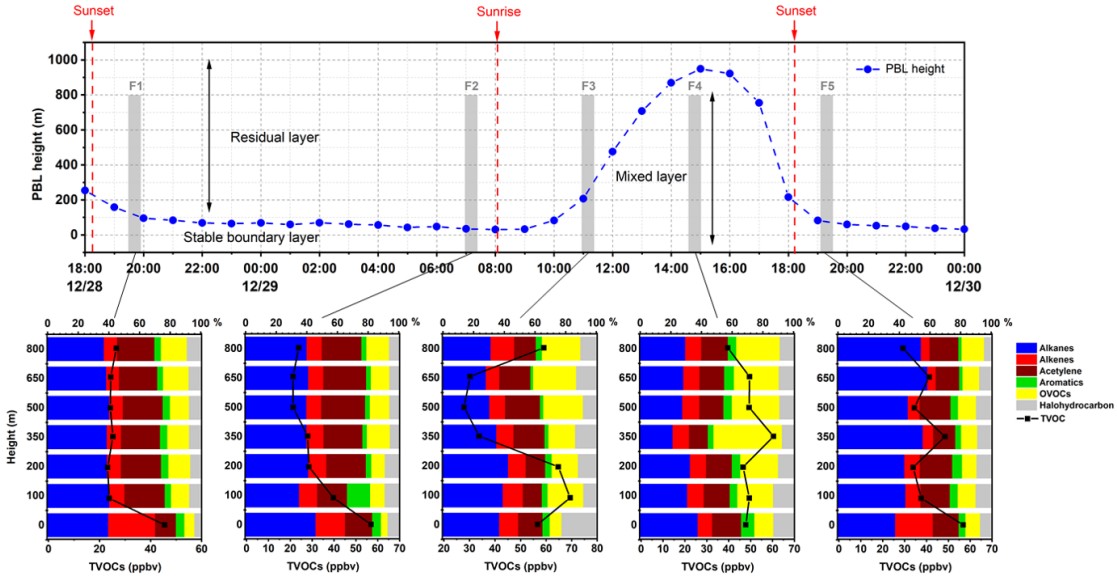

**Figure 6. Evolution of VOC vertical profiles with PBL development in five flights during 28–29 December (the grey area in the top panel is the time and height covered by each flight).**

Based on the obtained samples, we present vertical profiles of VOCs and other pollutants. To explore the overall regularity of the vertical evolution of VOCs, we calculated the average values of all the samples (Fig. S4). The vertical distribution of VOCs is a result of the combined effects of atmospheric physical diffusion and photochemical processes. Chemical decay in the

vertical direction is the major reason for the differences in the profiles of the different VOC groups. Except for OVOCs, the concentrations were the highest on the ground and decreased with altitude. From the surface to 800 m, alkanes, alkenes, acetylene, aromatics, and halohydrocarbons decreased by 38.4%, 61.7%, 27.9%, 55.7%, and 34.1%, respectively. Alkenes exhibited the fastest concentration reduction owing to their high reactivity. At all altitudes, inactive alkanes accounted for the largest proportion. The OVOC concentration increased with altitude owing to the secondary formation during the transmission

process. Meteorological factors also affected the vertical distribution of VOCs. We observed that the temperature inflection point corresponded to the maximum concentration during temperature inversion. Further analysis and scientific interpretation of these results and larger datasets will be the subject of a separate upcoming publication. The field experiment results demonstrated the application of VOC whole-air sampling equipment onboard the UAV for atmospheric vertical observation.

## 4 Conclusion

This study developed and characterised multi-channel whole-air sampling equipment onboard a UAV to investigate the vertical

distribution of VOCs in the PBL. The specially designed light-weighted sampling canisters were proved to be of good sampling integrity and storage stability. The integrated sampling equipment achieved multi-gradient near-continuous monitoring of VOCs and other parameters in the PBL, which is important for an in-depth understanding of the evolution of vertical profiles of VOCs during PBL development and the photochemical reaction progress in the upper atmosphere. This study extended the application of the UAV platform to atmospheric vertical observations, whose miniaturisation and manoeuvrability allowed high spatial-temporal resolution measurements. In future research, improving the endurance capability of UAV and integrating multi-functional measurement devices are worth exploring.

**Data availability**

The underlying research data can be accessed upon contact with the corresponding author (Xin Li: li_xin@pku.edu.cn).

**Author contribution**

XL and SY designed the study. SY analyzed the data and wrote the paper. XL, SY, YZ, DZ, HX, and SL contributed to the concept of UAV design. LZ, Ying Liu, SL, and YC contributed to the development of the quartz sampling canister. SY, XY, JC, and LW designed and performed the characterization experiment. SY, MS, LK, WW, and Yi Liu contributed to field observations and preprocessed data. XH, LW and XZ provided technical assistance locally. All authors contributed to the development of the manuscript.

**Competing interests**

The authors declare that they have no conflict of interest.

**Acknowledgements**

This work was supported by the National Natural Science Foundation of China (91844301) and by the Shenzhen Science and Technology Program (KCXFZ202002011006340).

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
