# Peer review of "Development of multi-channel whole-air sampling equipment onboard unmanned aerial vehicle for investigating VOCs vertical distribution in the planetary boundary layer"

_Atmospheric Measurement Techniques, 2022_

## Author Comment (AC1)

**Response to the Comments of the Reviewers**

Dear Editor and Reviewers,

We would like to thank you for the elaborate work on this manuscript.

We revised the manuscript by responding to each of the suggestions in the reviews. In our response, the questions of the reviewers are shown in *Italic* form and the responses in standard form.

We appreciate your help and time.

Sincerely yours,

Xin Li and Co-authors.

College of Environmental Sciences and Engineering
Peking University
100871 Beijing China
Email: li_xin@pku.edu.cn
Tel: +86-185 1358 6831
* * *
*Manuscript Number: amt-2022-277.*

*Manuscript Title: Development of multi-channel whole-air sampling equipment onboard unmanned aerial vehicle for investigating VOCs vertical distribution in the planetary boundary layer.*
* * *
**Response to Reviewer #1**

*General comments*

*Yang et al developed multi-channel whole-air sampling equipment onboard an unmanned aerial vehicle (UAV) platform, which is essential for measuring vertical concentrations of VOCs in the planetary boundary layer. The UAV platform is well designed and have been tested in field campaign in Chengdu city. The newly designed UAV successfully "capture" the characteristics of VOCs at different heights, demonstrating the capability of UAV for vertical VOC measurement. The manuscript is very well written. I recommend that the manuscript is published after some minor revisions.*

**Response:**

We would like to thank reviewer #1 for the positive and constructive comments, which helped us improve the quality of the paper. Below, we answer the reviewer's question point by point.
* * *
*Comments*

*1. Line 220, The impact of ozone on VOC concentration can be quantified for certain alkenes, if you have the reaction rate constant for a given alkene, the reaction time (7 days) and ozone concentration.*

**Response:**

We appreciate the reviewer's comments and agree with reviewer that the impact of $O_3$ on certain alkenes concentration can be quantified with the reaction rate constant, reaction time, and $O_3$ concentration. However, the $O_3$ concentration in the sampling canister could not be equal to the ambient concentration. If using the ambient $O_3$ concentration, the quantitative attenuation of alkenes will be far greater than reality. Taking ethene, propene, and 1,3-butadiene as examples, we used the box model to simulate the concentration change of alkenes and $O_3$ within a week (Fig. R1). The $O_3$ concentration in model was set to the ambient concentration during sampling (83 ppbv) and the reaction rate constant was taken from Goliff et al. (2013). Simulation results show that the 7-day recovery of ethene is only 15%, while propene and 1,3-butadiene almost disappear. This is significantly inconsistent with the measured 7-day recovery rate (red line). Therefore, the uncertainty of $O_3$ concentration in canisters makes it difficult to quantify the impact of $O_3$ on alkenes. Previous studies investigated the reasons of $O_3$ concentration uncertainty: 1. $O_3$ in canisters was destroyed by contact with the deactivated inner walls (Palluau et al., 2007a); 2. $O_3$ disappeared rapidly by auto-oxidation reactions (Harper, 2000); 3. $O_3$ was destroyed by deposition during the process of introducing into the canisters (Palluau et al., 2007b). However, it is certain that the residual ozone will react with alkenes, which is the reason that alkenes are the VOC group with the fastest decline different from other VOC species.

[Figure]

**Figure R1: Simulation of the alkenes and O₃ concentration during the reaction.**
* * *
*2. Line 292: Did you mean no significant emission sources are around the sampling site?*

**Response:**

Yes. We have clarified this sentence in the revised manuscript. Now the sentence reads as follows:

There were no significant emission sources are around the sampling site except for some motor vehicles.
* * *
*3. Line 62, 119, and 163, O3, NO2, SO2 -> O₃, NO₂, SO₂*

**Response:**

We appreciate the reviewer's comments, and we have revised accordingly in the manuscript.
* * *
**References**

Goliff, W. S., Stockwell, W. R., & Lawson, C. V. (2013). The regional atmospheric chemistry mechanism, version 2. Atmospheric Environment, 68, 174-185.

Harper, M. (2000). Sorbent trapping of volatile organic compounds from air. Journal of Chromatography A, 885(1-2), 129-151.

Palluau, F., Mirabel, P., & Millet, M. (2007a). Influence of ozone on the sampling and storage of volatile organic compounds in canisters. Environmental Chemistry Letters, 5(2), 51-55.

Palluau, F., Mirabel, P., & Millet, M. (2007b). Influence of relative humidity and ozone on the sampling of volatile organic compounds on Carbotrap/Carbosieve adsorbents. Environmental Monitoring and Assessment, 127(1-3), 177-187.

---

## Author Comment (AC2)

**Response to the Comments of the Reviewers**

Dear Editor and Reviewers,

We would like to thank you for the elaborate work on this manuscript.

We revised the manuscript by responding to each of the suggestions in the reviews. In our response, the questions of the reviewers are shown in *Italic* form and the responses in standard form.

We appreciate your help and time.

Sincerely yours,

Xin Li and Co-authors.

College of Environmental Sciences and Engineering
Peking University
100871 Beijing China
Email: li_xin@pku.edu.cn
Tel: +86-185 1358 6831
* * *
Manuscript Number: amt-2022-277.

Manuscript Title: *Development of multi-channel whole-air sampling equipment onboard unmanned aerial vehicle for investigating VOCs vertical distribution in the planetary boundary layer.*
* * *
**Response to Reviewer #2**

*General comments*

*The authors describe the design and initial tests of a new miniaturized whole air sampling system designed for a small unmanned aerial vehicle. They describe the design and procedures for collecting the WAS and compare the sampling efficiency to traditional electro-polished canisters. The results are of good use to the broader scientific community. The manuscript is well-written and should be published with only minor corrections.*

**Response:**

We would like to thank reviewer #2 for the positive and constructive comments, which helped us improve the quality of the paper. Below, we answer the reviewer's question point by point.
* * *
*Comments*

*1. The quartz samplers are the unique component of this system. I had a few more questions that could be addressed in the manuscript. Are the quartz samplers completely custom-made, or were they modified from commercially available glassware? If so, who manufactured or build the quartz samplers?*

**Response:**

We appreciate the reviewer's comments. The quartz sampler was designed by us and custom-made by the glass-maker workshop at Peking University. The quartz material was commercial one with $SiO_2$ purity of 99.99%. We made a clear statement of this in Line 98–99 in the revised manuscript:

As shown in Figure 1, this study designed and custom-made a new type of sampling canister composed of a quartz (99.99% purity of $SiO_2$) cylindrical tank and an airtight valve.
* * *
*2. It's unclear when and how the PTFE stopcock is unscrewed. I assume the stopcock is unscrewed manually after the sampler is connected to the solenoid valve. It doesn't state that there were any mechanical means to remotely open the PTFE stopcock while in flight. If that is the case, then please elaborate on those details.*

**Response:**

We apologize for the unclear presentation. Exactly as the reviewer assumed, the PTFE stopcock is unscrewed manually after the sampler is connected to the solenoid valve. We perform this operation before the UAV system takes off. All channels of the solenoid valve are closed at the initial state, ensuring the sampling canisters are isolated from the outside air even with the PTFE stopcock open. We carefully revised this part and added details in Line 109–114 in the revised manuscript. Now it reads as follows:

Before the UAV system took off, the PTFE stopcock was unscrewed manually after the sampler was connected to the solenoid valve. All channels of the solenoid valve were closed at the initial state, ensuring the sampling canisters were isolated from the outside air. The gas channel, where the blue cover is shown in Fig. 1, connected the sampling tube and solenoid valve. When sampling, the corresponding valve was opened, and ambient air was sucked into the canister through the channel.
* * *
*3. Is there a comparable stainless steel canister that you could include in comparing the weight/volume of the quartz samplers? How much weight are you saving if you installed small stainless steel samplers rather than quartz?*

**Response:**

We appreciate the reviewer's comments. We found a comparable commercial stainless steel canister (SilcoCan®) with a volume of about 1 L and a weight of about 1135 g. The quartz canister in this study has a volume of 1.1 L and a weight of 415 g. The weight of a single stainless steel canister is 2.7 times that of a quartz canister. Usually six canisters are carried in one flight mission. The installation of quartz canisters can save 4.32 kg in weight compared with stainless steel canisters, which is considerable. If other ancillary loads are unchanged, only 2 stainless steel canisters can be installed under the maximum payload limit. We added the following discussion in Line 99–103 in the revised manuscript:

The quartz canister was 14.3 cm high, 9.8 cm in diameter, 0.7 cm in wall thickness, approximately 1.1 L in volume, and approximately 415 g in weight. Benefiting from its light weight, six canister samples can be taken during one flight mission. There is a comparable commercial stainless steel canister (SilcoCan®) with a volume of 1 L and a weight of 1135 g, which is 2.7 times in weight of a quartz canister. The installation of quartz canisters can save 4.32 kg in weight compared with stainless steel canisters, which is considerable.
* * *
*4. It is common to add water vapor during the final stages of cleaning whole air samplers to help further passivate interior surfaces with a few monolayers of clean water. See references for U.S. EPA methods TO-15 as an example. Was that done here?*

**Response:**

We appreciate the reviewer's comments. We performed this operation in the process of purging the quartz canister with nitrogen. The Entech 3100D canister cleaning procedure includes the step of adding water vapor. We pour the distilled water into the water adding device behind the cleaning instrument and use high-purity nitrogen to purge the water. During the final stages of cleaning, we inject high-purity nitrogen passing through the water adding device into the canister. We have modified Line 116–119 in the revised manuscript. Now it reads as follows:

Before sampling, the quartz canister was purged with nitrogen and depressurized to a near-vacuum state (30 mTorr) using an Entech 3100D canister cleaning system. During the final stages of cleaning, we inject high-purity nitrogen passing through the water adding device into the canister to help further passivate interior surfaces with a few monolayers of clean water.